# Transcriptome Analysis Reveals Key Genes and Pathways Associated with the Regulation of Flowering Time in Cabbage (*Brassica oleracea* L. var. *capitata*)

**DOI:** 10.3390/plants12193413

**Published:** 2023-09-28

**Authors:** Jiao Wang, Bin Zhang, Huiling Guo, Li Chen, Fengqing Han, Chao Yan, Limei Yang, Mu Zhuang, Honghao Lv, Yong Wang, Jialei Ji, Yangyong Zhang

**Affiliations:** 1State Key Laboratory of Vegetable Biobreeding, Institute of Vegetables and Flowers, Chinese Academy of Agricultural Sciences, Beijing 100081, China; wangjiaoynl@163.com (J.W.); 13126720352@163.com (B.Z.); caas_ivf_guohl@163.com (H.G.); 18205480752@163.com (L.C.); feng857142@163.com (F.H.); yanglimei@caas.cn (L.Y.); zhuangmu@caas.cn (M.Z.); lvhonghao@caas.cn (H.L.); jijialei@caas.cn (J.J.); wangyong@caas.cn (Y.W.); 2College of Horticulture, Qingdao Agricultural University, Qingdao 266109, China; yanchao@qau.edu.cn

**Keywords:** cabbage, flowering time, transcriptomic, expression analysis

## Abstract

Flowering time is an important agronomic trait in cabbage (*Brassica oleracea* L. var. *capitata*), but the molecular regulatory mechanism underlying flowering time regulation in cabbage remains unclear. In this study, transcriptome analysis was performed using two sets of cabbage materials: (1) the early-flowering inbred line C491 (P_1_) and late-flowering inbred line B602 (P_2_), (2) the early-flowering individuals F2-B and late-flowering individuals F2-NB from the F_2_ population. The analysis revealed 9508 differentially expressed genes (DEGs) common to both C491_VS_ B602 and F2-B_VS_F2-NB. The Kyoto Encyclopedia of Genes and Genomes (KEGGs) analysis showed that plant hormone signal transduction and the MAPK signaling pathway were mainly enriched in up-regulated genes, and ribosome and DNA replication were mainly enriched in down-regulated genes. We identified 321 homologues of *Arabidopsis* flowering time genes (*Ft*) in cabbage. Among them, 25 DEGs (11 up-regulated and 14 down-regulated genes) were detected in the two comparison groups, and 12 gene expression patterns closely corresponded with the different flowering times in the two sets of materials. Two genes encoding MADS-box proteins, *Bo1g157450* (*BoSEP2-1*) and *Bo5g152700* (*BoSEP2-2*), showed significantly reduced expression in the late-flowering parent B602 compared with the early-flowering parent C491 via qRT-PCR analysis, which was consistent with the RNA-seq data. Next, the expression levels of *Bo1g157450* (*BoSEP2-1*) and *Bo5g152700* (*BoSEP2-2*) were analyzed in two other groups of early-flowering and late-flowering inbred lines, which showed that their expression patterns were consistent with those in the parents. Sequence analysis revealed that three and one SNPs between B602 and C491 were identified in *Bo1g157450* (*BoSEP2-1*) and *Bo5g152700* (*BoSEP2-2*), respectively. Therefore, *BoSEP2-1* and *BoSEP2-2* were designated as candidates for flowering time regulation through a potential new regulatory pathway. These results provide new insights into the molecular mechanisms underlying flowering time regulation in cabbage.

## 1. Introduction

Cabbage (*Brassica oleracea* L. var. *capitata*) originated in Europe and is an important *Brassica* vegetable crop that is widely grown all over the world [1]. Flowering time determines the transition from the vegetative to reproductive stage of plants, and appropriate flowering time is crucial for plant growth and reproduction in a given environment [2]. Bolting and flowering at the right time helps plants adapt to conditions that increase seed and fruit production [3], while premature bolting seriously affects cabbage yield and quality [4,5,6]. Therefore, flowering time is an important characteristic in cabbage breeding.

Flowering is a complex process that is regulated by both environmental and internal signals, which are brought together by flowering integration factors to jointly determine whether to flower [7,8,9]. Previous studies have shown that there are at least six flowering regulation pathways in *Arabidopsis thaliana*: the photoperiod pathway, vernalization pathway, ambient temperature pathway, autonomic pathway, gibberellin pathway, and age pathway [10]. The plant’s response to the length of daylight is called photoperiodism [11].In the regulatory pathway of flowering time in the model plant *Arabidopsis thaliana*, the photoperiod pathway is one of the most well-studied and complete pathways [12,13]. The photoperiod pathway primarily perceives and responds to light signals through photoreceptors and the circadian clock, promoting the transition to flowering [14]. Known photoreceptors in plants are mainly divided into three classes, namely phytochromes (including *PHYA*, *B*, *C*, *D*, and *E*), cryptochromes (including *CRY1*, *CRY2*, and *CRY3*), and phototropins (composed of *phot1 1* and *phot1 2*) [15]. Photoreceptors are responsible for transmitting light signals to the circadian clock, which generates rhythmic oscillation signals [13]. *CO* (*CONSTANS*) is a central regulatory factor in the photoperiod pathway [16], which can induce the expression of *FT* (*FLOWERING LOCUS T*) and promote flowering [14]. The phenomenon in which winter plants undergo a period of prolonged low temperature to transition from the vegetative growth phase to the reproductive growth phase is known as vernalization [17]. Notably, the cold memory acquired by plants experiencing long-term low temperature is not inherited by the next generation [18]. In the model plant *Arabidopsis thaliana*, whether the plants have a flowering need for vernalization is predominantly governed by two genes, *FRI* (*FLOWERING LOCUS C-INTERACTING PROTEIN 1*) and *FLC* (*FLOWERING LOCUS C*) [19]. FRI is a coiled helix protein, part of the superprotein complex, a transcriptional activator of *FLC* that increases mRNA transcription levels [20,21]. *FLC*, a member of the MADS-box family, functions as a key regulator in the repression of flowering and is predominantly expressed in the shoot apical meristem. It exerts its inhibitory effect on flowering by negatively regulating the expression of *SOC1*/*AGL20* (*SUPPRESSOR OF OVEREXPRESSION OF CONSTANS 1*/*AGAMOUS-LIKE 20*) and *FT*, which are known as flowering promoting factors [22]. Furthermore, the PRC2 complex plays a pivotal role in the vernalization pathway by modulating the transcriptional repression of *FLC* [23]. *FLC*, as a star gene in the regulation of the flowering time pathway, has been widely studied. Recently, two additional pathways, namely the environmental temperature pathway and the age pathway, have been proposed, and the inclusion of these pathways further enriches our understanding of the complex regulatory network governing the intricate mechanisms of flowering in plants. The environmental temperature pathway is a mechanism used by plants to perceive and respond to changes in environmental temperature, thereby regulating the timing of flowering. The *SVP* (*SHORT VEGETATIVE PHASE*) gene plays an important role in this pathway [10]. Like *FLC*, *SVP* is also a flowering repressor factor in the MADS-box family. Under low-temperature conditions, *SVP* is activated and expressed, which inhibits the expression of *FT* and consequently suppresses flowering. Conversely, under high-temperature conditions, the expression of *SVP* is inhibited, allowing *FT* to be expressed [24]. Therefore, *Arabidopsis* is more likely to flower in high-temperature environments under normal circumstances. In the age pathway, the timing of flowering in plants is intricately linked to their individual growth duration and developmental stage [9]. Within this pathway, genes such as *SPL*s, miR156, and miR172 assume pivotal roles, exerting a significant influence on the regulation of flowering [10,25,26,27]. The autonomous pathway, a remarkable phenomenon in plant biology, enables plants to trigger flowering independently of external environmental cues by harnessing their own internal factors. Within this intricate pathway, a multitude of genes, including *FCA* (*FLOWERING CONTROL LOCUS A*), *FY*, *FLD* (*FLOWERING LOCUS D*), *FVE*, *COOLDAIR*, and others, have been meticulously identified [28]. These genes play pivotal roles by intricately modulating the transcription and expression of *FLC* through histone modification, thereby orchestrating the delicate process of flowering in plants [29]. Plant hormones are essential for regulating plant growth and development. Among various plant hormones, gibberellin (GA) plays a major role in determining flowering time [30]. The gibberellin pathway exerts a pivotal influence on the intricate process of plant flowering, primarily through the intricate interplay between the *GID1* (*GA INSENSITIVE DWARF 1*) gene and the DELLA protein, in which the DELLA protein plays a crucial role. The DELLA protein, especially, serves as a master regulator, orchestrating the precise timing and progression of the flowering process [31,32]. Furthermore, the GA pathway intricately interacts with a multitude of other genetic pathways governing flowering, as well as plant hormone signaling pathways, thereby establishing a complex network of regulatory crosstalk that ensures the harmonious coordination of plant development [32]. The photoperiod and vernalization pathways control flowering in response to seasonal changes in day length and temperature. The environmental temperature pathway responds to daily growth temperature. On the other hand, the age, autonomous, and gibberellin pathways act more independently of environmental stimuli. These six pathways are independent of each other and interact with each other [10]. Together, these pathways form a complex network that integrates various environmental and endogenous signals to precisely control the timing of flowering in plants.

In recent years, transcriptome analysis has been performed on many plant species using RNA-seq to reveal the molecular mechanisms governing the regulation of flowering time. Song et al. (2017) identified 19 differentially expressed genes (DEGs) associated with flowering time in maize via transcriptome analysis [33]. Wang et al. (2021) conducted transcriptome analysis using the late-flowering Moneymaker tomato cultivar and the earlier-flowering 20965 cultivar, which uncovered 49 candidate genes related to flowering time in tomato [34]. Wei et al. (2021) performed transcriptome analysis using late-bolting (Y410-1 and SY2004) heading Chinese cabbage and early bolting Chinese cabbage (CX141), which revealed that six DEGs related to phytohormones and signal transduction were up-regulated in late-flowering Chinese cabbage lines (Y410-1 and SY2004) and were therefore considered candidate genes for the regulation of flowering time in *B. rapa* [35]. Through QTL mapping and transcriptome analysis, Jian et al. (2019) identified eight genes as candidates for the regulation of flowering time in *B. napus,* including *PSEUDO RESPONSE REGULATOR 7* (*PRR7*) and *FY* [36]. Through transcriptome analysis and QTL mapping, Liu et al. (2022) identified six candidate genes that are involved in the photoperiod pathway, in the vernalization pathway, and in plant hormone signaling that affect flowering time in *B. napus* [37].

In higher plants, flowering is a complex process controlled by multiple genes. *FLC* is a strong inhibitory factor of flowering [37,38], and its high expression can delay flowering [39,40]. Li et al. (2022) found that a 215 bp deletion in the intron of *BoFLC2* slows its silencing activity by feeding back to the core gene of the PHD-PRC2 complex, which was the main cause of delayed flowering [41]. Castillejo et al. (2008) identified the *TEMPRANILLO* genes (*TEM1* and *TEM2*) as novel flowering suppressors that delay flowering time by inhibiting *FT* expression [42]. *TEM1* is a transcriptional repressor that negatively regulates flowering in *Arabidopsis thaliana* [43]. Early-flowering is induced by the overexpression of *IiSEP2* in *Arabidopsis thaliana* [44]. The overexpression of miR-156 downregulates the *SPL* gene and delays flowering time in *Arabidopsis thaliana*, which has 16 members of the *SPL* family [45]. The overexpression of *BrpSPL9* in Chinese cabbage leads to premature flowering [46]. The *LATE ELONGATED HYPOCOTYL* (*LHY*) gene encodes a circadian clock protein, and *lhy* mutants showed a late-flowering time under a long light period in *Arabidopsis thaliana* [47].

C491 is a late-flowering winter cabbage inbred line, and B602 is an early-flowering winter cabbage inbred line. There was no significant difference in other traits, except flowering time between the two lines. In this study, we generated a F_2_ segregating population using C491 (P_1_) and B602 (P_2_). Field investigation showed that flowering time was a quantitative trait in the F_2_ population (Appendix A). We performed transcriptome analysis using the early-flowering inbred line C491 (P_1_) and late-flowering inbred line B602 (P_2_), and the early-flowering line F2-B and late-flowering line F2-NB from the F_2_ population in cabbage, with a focus on genes related to the regulation of flowering time. The findings of this study provide theoretical support for studying the molecular mechanism, candidate genes, and breeding of late-flowering time in cabbage.

## 2. Results

### 2.1. RNA-Seq and DEG Analysis of C491, B602, F2-NB, and F2-B Leaves

Twelve cDNA libraries for C491 (Figure 1a), B602 (Figure 1b), F2-NB, and F2-B were sequenced to identify differentially expressed genes (DEGs) that are potentially involved in regulating flowering time. After removing the adaptor sequence, contamination sequence, and low-quality reads, 88.97 million (C491), 63.93 million (B602), 125.78 million (F2-NB), and 147.13 million (F2-B) clean reads were recovered, and the Q30 of all reads was higher than 93% (Appendix A). The clean reads were mapped to the *B. oleracea* TO1000 reference genome (http://plants.ensembl.org/Brassica_oleracea/Info/Index, accessed on 1 May 2020), with all mapping rates being greater than 62.5%. The density distributions and boxplots of all genes showed the same pattern in parents and offspring, respectively (Appendix A).

In the comparison groups C491_VS_B602 and F2-B_VS_F2-NB, a total of 9137 and 9508 DEGs were detected, respectively. In C491_VS_B602, 4046 up-regulated genes and 5091 down-regulated genes were identified. There were 4219 down-regulated genes and 5289 up-regulated genes identified in F2-B_VS_F2-NB (Appendix A). Then, we identified 3748 identical DEGs in the two groups, including 1710 up-regulated genes (logFC > 1) and 2038 down-regulated genes (logFC < −1) (Figure 2a).

### 2.2. GO and KEGG Enrichment Analysis of Overlapping DEGs

Gene Ontology (GO) analysis was performed on the up-regulated and down-regulated genes, and all the DEGs were mapped to the terms of the GO database for annotation. The DEGs were classified into three GO categories: biological process (BP), cellular component (CC), and molecular function (MF). The top 20 enriched terms were identified in each comparison group. The response to stimulus (BP) and response to hormone (BP) were mainly enriched in up-regulated genes, while the cell cycle (BP) and cell cycle process (BP) were mainly enriched in down-regulated genes (Figure 2b, Appendix A).

The Kyoto Encyclopedia of Genes and Genomes (KEGGs) analysis was subsequently performed to uncover the important biological functions of the DEGs, which resulted in 3748 DEGs being grouped into 33 KEGG pathways (10 KEGG pathways were enriched in up-regulated genes, and 23 KEGG pathways were enriched in down-regulated genes). The KEGGs analysis indicated that plant hormone signal transduction, the MAPK signaling pathway, and plant-pathogen interaction were mainly enriched in up-regulated genes, while ribosome, DNA replication, glyoxylate, and dicarboxylate metabolism were the main pathways enriched in down-regulated genes (Figure 3c, Appendix A).

### 2.3. Identification of DEGs Involved in the Flowering Time Regulation Pathway

To identify flowering time genes (*Ft*) in the overlapping genes of the two groups, we used 147 flowering time genes identified in *Arabidopsis* for a Basic Local Alignment Search Tool (BLAST) search in cabbage [48]. Of 147 putative *Ft* genes known in *Arabidopsis*, a BLAST search identified 321 *Ft* homologous genes in cabbage (Appendix A). The *Ft* genes were classified into major flowering pathways as follows: ‘C’ (circadian clock pathway) (152 genes); ‘L’ (light signaling pathway) (152 genes); ‘P’ (photoperiod pathway) (152 genes); ‘V’ (vernalization pathway) (62 genes); ‘A’ (autonomous pathway) (36 genes); ‘G/M’ (gibberellin signaling and metabolism) (32 genes); ‘D/M’ (development and metabolism response) (32 genes); ‘I’ (integrator) (5 genes); and ‘A’ (age pathway) (2 genes) [49]. Jian et al. used RNA-seq analysis to detect 105 DEGs related to flowering time in *B*. *napus* lines with extreme differences in flowering time [36]. Song et al. identified 175 flowering-time-related *Arabidopsis* homologues in *B. napus* via RNA-seq analysis, of which 91 were involved in the photoperiodic pathway, 48 in the autonomous pathway, and 32 in the vernalization pathway [5]. We speculate that the photoperiodic pathway, vernalization pathway, and autonomous pathway may be the key pathways regulating flowering time in cabbage.

Among the overlapping up-regulated and down-regulated genes of C491_VS_B602 and F2-B_VS_F2-NB, 25 genes (11 up-regulated and 14 down-regulated genes) were related to flowering (Appendix A). Among the 25 genes, 10 (40%) were related to the photoperiodic pathway, 8 (32%) to plant growth and development, 3 (12%) to the vernalization pathway, 2 (8%) to the age pathway, and 1 (4%) to the GA pathway and autonomic pathway (Appendix A).

### 2.4. qRT-PCR Validation of Flowering-Related DEGs

The expression patterns of 25 flowering-related genes were verified via qRT-PCR (Figure 3) and were consistent with the RNA-seq data (Figure 4). Among them, 12 gene expression patterns, such as *BoSEP2-1*, *BoSEP2-2,* and *BoTPS1*, closely corresponded with flowering time traits. *Bo5g152700* (*BoSEP2-1*) and *Bo1g157450* (*BoSEP2-2*), encoding MADS-box proteins, showed significantly reduced expression in late-flowering B602 compared with early-flowering C491 using qRT-PCR analysis, which was consistent with the RNA-seq data. The expression patterns of *Bo5g152700* (*BoSEP2-1*) and *Bo1g157450* (*BoSEP2-2*) were investigated in two additional sets of early-flowering and late-flowering materials (early-flowering NCF8-1 and late-flowering NCF11-1; early-flowering HC101-1 and late-flowering HB9-1). Our findings revealed that the expression patterns of these two genes in the additional early-flowering and late-flowering materials were in line with those observed in the parental lines C491 and B602 (Appendix A). Sequence analysis revealed that two SNPs were identified at the 17 bp (T-A) and 18 bp (G-A) on the CDS of *Bo5g152700* (*BoSEP2-1*) in B602 compared with C491, resulting in the change of amino acid V to E; and a synonymous mutation was identified at the 232 bp (G-T) on the CDS of *Bo5g152700* (*BoSEP2-1*) in B602 compared with C491. In addition, a nonsynonymous mutation was identified at the 277 bp (A–G/R-G) on the CDS of *Bo1g157450* (*BoSEP2-2*) in B602 compared with C491 (Figure 5). In conclusion, we speculated that *BoSEP2-1* and *BoSEP2-2* were strong candidate genes for regulating flowering time in cabbage.

## 3. Discussion

In *B*. *napus*, Jian et al. (2019) identified 9169 DEGs, including 105 genes related to flowering time. KEGGs enrichment analysis revealed that phenylpropionic biosynthetic transporters and plant hormone signal transduction were the pathways significantly enriched in these DEGs [36]. In *B. napus*, Song et al. (2021) identified 11,888 DEGs in which metabolism and biosynthesis of other secondary metabolites were significantly enriched [5]. In *B. rapa*, Wei et al. (2021) identified 22,396 DEGs in three transcriptome groups. The KEGGs analysis showed that plant hormone and signal transduction and starch and sucrose metabolism were the processes in which these DEGs were most enriched [35]. In this study, a total of 3748 common DEGs were identified in C491_VS_ B602 and F2-B_VS_ F2-NB. The KEGGs analysis showed that ribosomes, DNA replication, plant hormone signal transduction, and the MAPK signaling pathway were most enriched in these DEGs. Our results differ from those of previous studies, revealing the complexity of the flowering time regulatory network.

As one of the important links in plant growth and development, the factors regulating plant flowering time have received extensive attention. According to previous reports, there are many transcription factors in plants that can affect the time of flowering. Wang et al. (2014) showed that the interaction of *CO* and *BBX19* could inhibit *Arabidopsis* flowering [50]; Zhang et al. (2016) found that the transcription factor *TaMYB72* promotes flowering in rice [51]; Zhou et al. (2018) found that the GRAS transcription factor *SlGRAS26* in tomato plants can regulate the flowering time of plants [52]; Yao et al. (2019) found that *SPL10* can influence the flowering time of *Arabidopsis thaliana* [2]; Hong et al. (2021) found that *MYB117* could negatively regulate the flowering time of *Arabidopsis thaliana* [53]. In this study, we obtained 25 flowering-time-related transcription factors through transcriptome analysis, including 12 transcription factors that positively regulate-flowering time and 13 transcription factors that negatively regulate-flowering time.

Shu et al. (2018) identified *Bol024659* (*BolGRF6*) as a candidate gene for the regulation of flowering time in broccoli and cabbage via transcriptome analysis [54]. Shah et al. (2018) used QTL mapping for transcriptome analysis and identified 36 homologous genes of flowering-time-related genes, such as *Bna. FLC* and *Bna. AGL19* in *B. napus* as candidate genes [55]. Song et al. (2021) identified 175 DEGs using transcriptome analysis, combining the analysis with QTL mapping, and *BnaA03g03410D* (*EMF1*), *BnaA03g04040D* (*NF-YA1*), *BnaA03g30130D* (*COL9*), and *BnaA03g22100D* (*TOE1*) were identified as candidate genes that regulate-flowering time in *B. napus* [5]. Zheng et al. (2022) used QTL mapping and transcriptome analyses in *Elymus sibiricus* to identify 72 candidate genes, including *LATE*, *GA2OX6*, *FAR3*, and *MFT1* [56]. In contrast to the findings of other studies, 25 flowering-time-related DEGs were identified in this study, among which *Bo1g157450* (*BoSEP2-1*) and *Bo5g152700* (*BoSEP2-2*) were significantly down-regulated in bolt-resistant materials, so these two genes were selected as the candidate genes. These results indicate that flowering time is a complex process.

In woodland strawberry and *Arabidopsis*, the overexpression of *FvWRKY71* resulted in early-flowering, and the expression of *SEP2* was found to be up-regulated in *Isatis indigotica*, which may be responsible for flowering time regulation [57]. Ma et al. (2022) cloned *liSEP2* in Woad (*Isatis indigotica*) and overexpressed *LiSEP2* in *Arabidopsis thaliana*, resulting in early-flowering and reduced flowers and lateral branches [44]. Myat et al. (2022) discovered that the significantly down-regulated expression of *SEP1* and *SEP2* resulted in delayed flowering in cotton [58]. In this study, the expression of both *BoSEP2-1* and *BoSEP2-2* was significantly down-regulated in late-flowering B602, which was similar to the results of the above studies, further indicating the positive regulation of flowering time by *SEP2*. The above results lay the foundation for further studies on the molecular mechanism of flowering time regulation in cabbage.

In summary, based on transcriptome data analysis, we proposed a new regulatory network for the regulation of flowering time in cabbage (Figure 6). The regulation of flowering time using *SEP2* may be realized through the original flowering time pathway. In addition, *SEP2* may regulate-flowering time independently of the original flowering time pathway. The specific interaction mechanism needs to be further studied.

## 4. Conclusions

In this study, two MADS-box genes, *Bo1g157450* (*BoSEP2-1*) and *Bo5g152700* (*BoSEP2-2*), were identified as candidates for flowering time regulation using transcriptome analysis and qRT-PCR validation. This study lays a foundation for revealing the molecular mechanisms underlying flowering time regulation in cabbage.

## 5. Materials and Methods

### 5.1. Plant Materials

C491 (P_1_) is an early-flowering cabbage inbred line, and B602 (P_2_) is a late-flowering cabbage inbred line. F2-NB1, F2-NB2, and F2-NB3 are late-flowering lines, and F2-B1, F2-B2, and F2-B3 are early-flowering lines derived from the F_2_ population constructed by B602 and C491. When the early-flowering materials C491 and F2-Bs open the first flower (while the late-flowering materials have not yet blossomed), young bolt leaves of the early-flowering materials and heart leaves of the late-flowering materials were taken for constructing cDNA libraries [5]. The other materials, NCF8-1(early-flowering), NCF11-1(late-flowering), HC101-1 (early-flowering), and HB9-1 (late-flowering), were used to verify that the expression patterns of the candidate genes were the same as those of the parents, and these four materials were consistent with the sampling method of the parents. Each sample was collected from three individuals. All materials were provided by the Institute of Vegetables and Flowers, Chinese Academy of Agricultural Sciences, and were planted in a greenhouse (25 °C ± 2 °C) under a 16 h light/8 h dark photoperiod at the Shouguang Vegetable Research and Development Center in Shandong on 16 September 2022. Colonization was performed using the single-row colonization method, with a row spacing of 80 cm and a plant spacing of 45 cm.

### 5.2. RNA Extraction and Sequencing

Total RNA from all of the collected samples was extracted using a Tiangen RNAprep Pure Plant Kit (Tiangen Biotechnology Co., Ltd., Beijing, China) according to the manufacturer’s instructions. RNA purity and quality were determined using a spectrophotometer (BioDrop, Cambridge, UK) and agarose gel electrophoresis. First-strand cDNA was synthesized using the TIANGEN FastKing RT Kit following the manufacturer’s instructions. A total of 12 cDNA libraries were constructed and subsequently sequenced using the Biomarker Technologies Co. Illumina HiSeq 2000 platform, Ltd. (Beijing, China).

### 5.3. Data Analysis

Using HISAT [59,60], clean reads were aligned to the *Brassica oleracea* TO1000 reference genome (http://plants.ensembl.org/Brassica_oleracea/Info/Index (accessed on 26 January 2023). Using the selection criteria |log2 (fold-change)| > 1 and q-value ≤ 0.05 for significant differential expression, DEGseq was used to identify DEGs. Using cluster Profiler software, Gene Ontology (GO) functional enrichment analysis and Kyoto Encyclopedia of Genes and Genomes (KEGGs) pathway enrichment analysis were performed on the DEGs.

### 5.4. qRT–PCR Validation

First-strand cDNA was synthesized using the TIANGEN FastKing RT Kit. qRT-PCR was carried out using a TransStart Top Green qPCR SuperMix Kit (TransGen Biotech, Beijing, China) on a CFX96 Real-Time System (Bio-Rad, Hercules, CA, USA). All experiments were performed with three biological and four technical replicates. The relative expression levels of the genes were calculated using the 2^−∆∆Ct^ method [61]. *B. oleracea* actin was used as the internal reference gene. The qRT-PCR primers used are shown in Appendix A.

## Figures and Tables

**Figure 1 plants-12-03413-f001:**
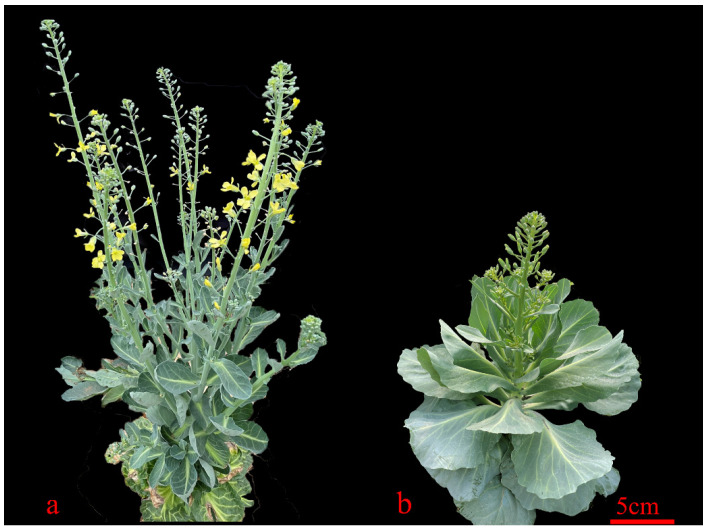
Phenotypes of C491 and B602 at 41 days after vernalization. (**a**) C491 is early-flowering. (**b**) B602 is late-flowering. Bar = 5 cm.

**Figure 2 plants-12-03413-f002:**
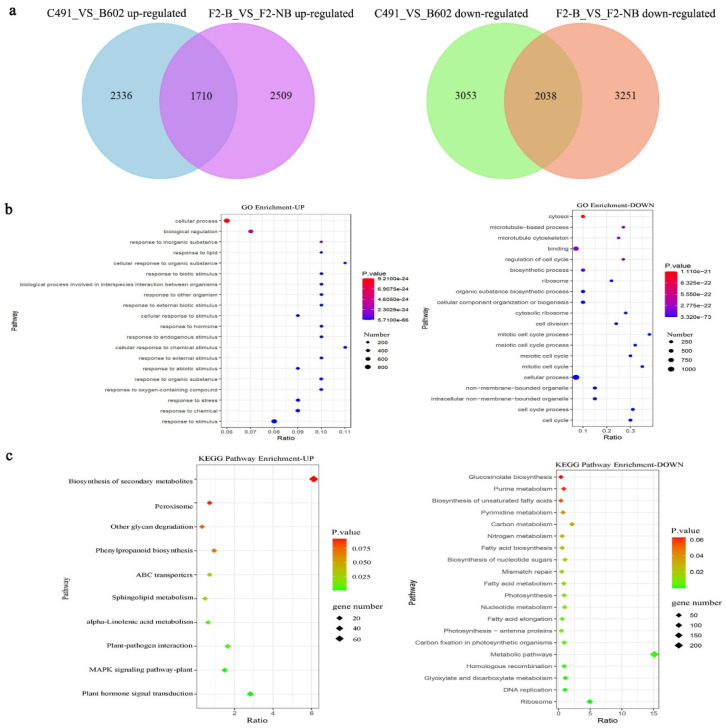
Comparison of differentially expressed genes response to vernalization between C491_VS_B602 and F2-B_VS_F2-NB. (**a**) Venn diagram showing the numbers of up- and down-regulated overlapping DEGs in C491_VS_B602 and F2-B_VS_F2-NB, respectively. (**b**) Top 20 enriched GO terms for up- and down-regulated overlapping DEGs in both C491_VS_B602 and F2-B_VS_F2-NB. (**c**) Top 20 enriched KEGG pathways for up- and down-regulated DEGs in both C491_VS_B602 and F2-B_VS_F2-NB.

**Figure 3 plants-12-03413-f003:**
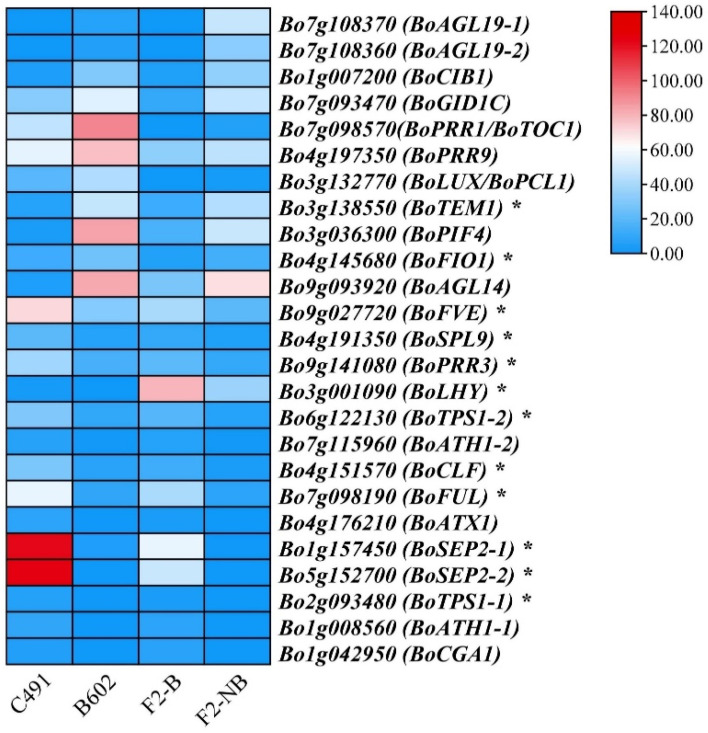
Heatmap showing the gene expression of 25 flowering-related DEGs. The intensity of the color for each gene model corresponds to the average fragments per kilobase of transcript per million mapped reads (FPKM) values. The asterisks represent significant differences (*p* < 0.01).

**Figure 4 plants-12-03413-f004:**
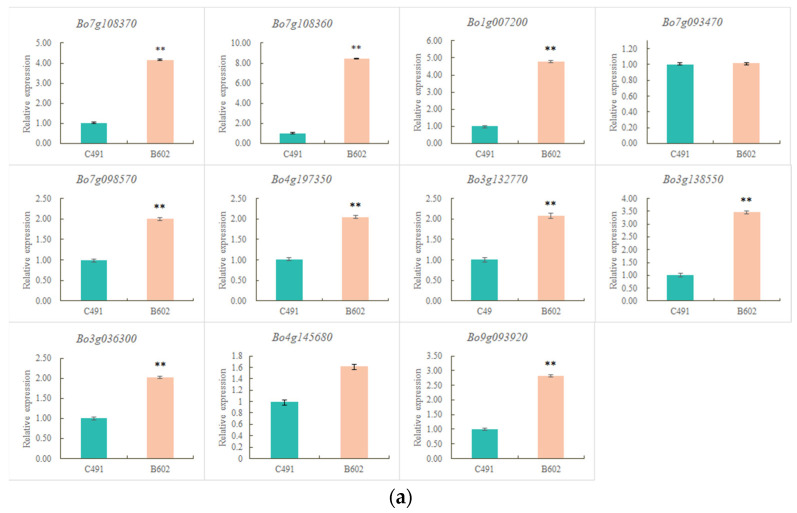
Validation of 25 DEGs related to flowering time via qRT-PCR. (**a**) The relative expression levels of 11 up-regulated DEGs related to flowering time in early-flowering C491 and late-flowering B602. (**b**) The relative expression levels of 14 down-regulated DEGs related to flowering time in early-flowering C491 and late-flowering B602. ** indicates a significant difference (*p* < 0.01).

**Figure 5 plants-12-03413-f005:**
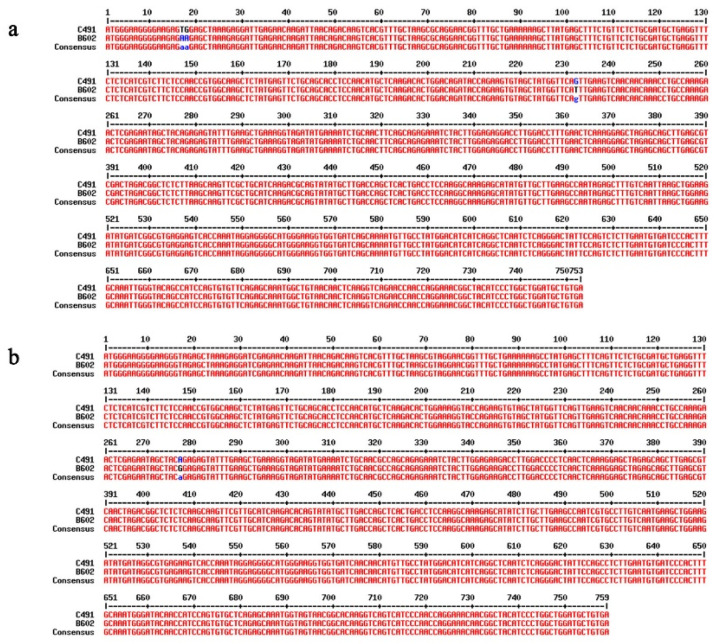
CDS sequence difference of the genes *Bo5g152700* (*BoSEP2-1*) and *Bo1g157450* (*BoSEP2-2*) between C491 and B602. (**a**) CDS sequence difference of the gene *Bo5g152700* (*BoSEP2-1*) between C491 and B602. (**b**) CDS sequence difference of the gene *Bo1g157450* (*BoSEP2-2*) between C491 and B602. The red bases represent consistency between the parents, while the blue and black bases represent sequence differences between the parents.

**Figure 6 plants-12-03413-f006:**
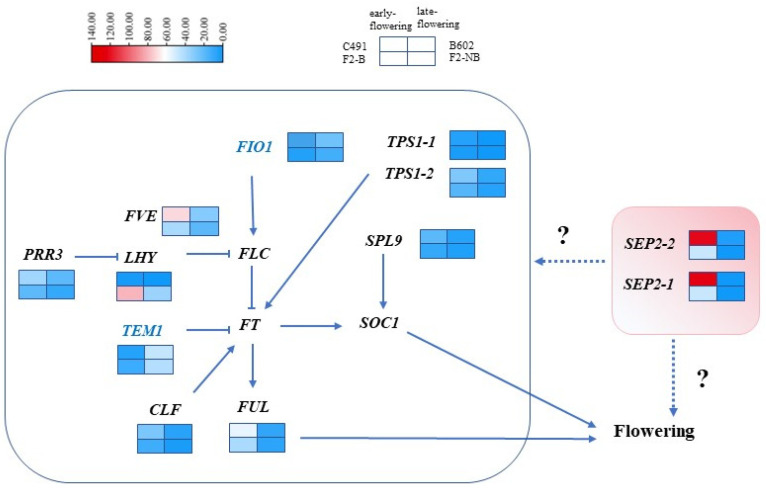
Potential flowering time regulatory networks in cabbage. Genes in blue represent flowering repressors, and genes in black represent flowering promoters. The genes in the black box have regulatory interactions that have been reported before, and the genes in the pink box are candidates. The arrows indicate transcriptional activation, whereas the bars indicate transcriptional repression. The dashed line represents the putative regulatory pathway.

## Data Availability

All the data generated or analyzed in this study are included in this published article and its Appendix A.

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
