# Peer review of "Transcriptome Analysis Reveals Key Genes and Pathways Associated with the Regulation of Flowering Time in Cabbage (*Brassica oleracea* L. var. *capitata*)"

_plants, 2023, doi:10.3390/plants12193413_

Round 1

Reviewer 1 Report

Wang et al., realized transcriptomic analysis from two sets of cabbage: (1) the early-flowering inbred line C491 (P1) and late-flowering inbred line B602 (P2), (2) the 15 early-flowering individuals F2-B and late-flowering individuals F2-NB from the F2 population. They found 9508 DEGs common C491 vs B602; F2-B vs F2 NB., and KEGG analysis showed that plant hormone signaling, MAPKs pathway, ribosome, and DNA replication were enriched. Among them, two gene-encoding MADS-box proteins, BoSEP2-1 and BoSEP2-2 showed significantly reduced expression, suggesting strong candidates for flowering time regulation. I found interesting the paper and I think that this research could contribute to the knowledge that we have in these plant species. However, I have some comments.

Major comments:

The results only showed transcriptomic analysis and predictions for signaling networks.

I consider that the results are prelaminar because the authors do not show any genetic evidence. It will be interesting if authors generate a KO or miRNA line in BoSEP2-1 or BoSEP2-2 genes and analyzed the flowering time in these lines. Also, the authors could analyze more traits associated with flowering time, or senescence symptoms, including leaf number, floral stem length, etc. 

Minor comments:

The authors should indicate which part of the plant is used for RNA extraction and RNA-seq analysis, as well as the plant old. It is very important because each region/time affects the transcriptomic finger. 

Reviewer 2 Report

Some concerns about this paper:

1) Line 64-66, authors misunderstood the original paper [24], please check it again;

2) What is the age of plants in Fig. 1? Please add such information in the figure legend;

3) line 223, section 4.1, what is the age of your plant materials used for sequencing? Is there any references guiding you to use young leaves to explore flowering time in cabbage?

4) line 236, typos;

5) line 230-231, please provide details of libraries construction;

6) lines 130-135, how did authors classify the 321 genes, is there any references?

7) line 156-157, how about Bo7g093470, Bo3g036300, Bo9g093920, they could be enhancers of flowering? Why can't they be strong candidate regulating flowering time?

8) Is there any other different phenotypes of C491 versus B602, and F2_B versus F2_NB except for the flowering time? If the answer is no, please introduce their backgrounds in the Introduction Section, especially for C491 and B602. If the answer is yes, exploring the 3748 common DEGs in C491 VS B602 and F2-B VS F2-NB to screen candidate genes of flowering time is not appropriate.

9) The Discussion Section is not enough.

Reviewer 3 Report

Where are the references in the material and methods part?

Why did you measure and take the samples from leaves? why did not investigate other parts?

Round 2

Reviewer 1 Report

I belive that the manuscript lacks of genetic evidence to validate the transcriptomic study. "TranscriptomeAnalysis Reveals Key Genes and Pathways Associated with the Regulation of Flowering Time in Cabbage (Brassica oleracea L. var. capitata)". 

The author realized a trancriptomic analysis in two cabbage species, (1) the early-flowering inbred line C491 and the late-flowering inbred line B602. The authors identified two potential candidates for regulating flowering time, the MADS-box BoSEP2-1 and BoSEP2-2. This is an interesting study, however I consider that it is very important genetic validation to be considered in Plants.

1.- The study only shows a transcriptomic analysis. From it, the authors identified the DEGs (Fig. 3), realyzed a GO analysis (Figure 2), showed a heat map (Fig. 4), realized RT-qPCR as a control to validate the transcriptomic analysis (Fig. 5), and generated a regulatory network from the same analysis (Fig. 6). In other reports, the transcriptomic analysis is shown in one figure, whereas in this report the authors generated multiple analyses/figures from the same transcriptomic study.  

In my previous comments, I suggested to authors that the transcriptomic study should be validated by genetic information, in particular the two genes identified as potential regulators for flowering time. I suggested that the authors generate a Cabbage transgenic which could over express BoSEP2-1 or BoSEP2-2, or generate cabbage plants with reduced expression of BoSEP2-1 or BoSEP2-2, like a micRNA. Without this information I can not suggest the publication in Plants, maybe in another Journal specialized in genomics.  

Author Response

Response: Thanks for the comments. We have merged picture 3 into picture 2,  line 174.

The functional verification of some genes in cabbage is limited by varieties and genotypes. We have tried to overexpress the BoSEP2-1 and BoSEP2-2 genes in late-flowering B602 and knockout the BoSEP2-1 and BoSEP2-2 genes in early-flowering C491, but failed. And we are still trying to verify the BoSEP2-1 and BoSEP2-2 genes function in different cabbage varieties. We have added information on the sequence differences between C491 and B602 of the candidate genes in lines 228-235.

Reviewer 2 Report

No more comment

Author Response

Thank you for your review.

Round 3

Reviewer 1 Report

I found that the authors included interesting additional information related to BoSEP2-1 and BoSEP2-2. If is very difficult to generate an transgenic plant in cabbage, It could interesting analyze the expression levels of BoSEP1 and BoSEP2 in other varieties with late and/or early flowering time, and compare the results with C491 and B602. I think that it would strength the main conclusion. 

Author Response

Thanks for the comments. We have added the results of qRT-PCR, lines 235-241.